# Are there inequalities in the attendance and effectiveness of behavioural weight management interventions for adults in the UK? Protocol for an individual participant data (IPD) meta-analysis

Jack M Birch ![ORCID],[1] Julia Mueller ![ORCID],[1] Stephen Sharp,[1] Jennifer Logue ![ORCID],[2] Michael P Kelly,[3] Simon J Griffin,[1,3] Amy Ahern[1]

¹MRC Epidemiology Unit, University of Cambridge, Cambridge, UK
²Lancaster Medical School, Lancaster University, Lancaster, UK
³Department of Public Health and Primary Care, University of Cambridge, Cambridge, UK

**Correspondence to**
Jack M Birch;
jack.birch@mrc-epid.cam.ac.uk

## ABSTRACT

**Introduction** It is important to identify whether behavioural weight management interventions work well across different groups in the population so health inequalities in obesity are not widened. Previous systematic reviews of inequalities in the attendance and effectiveness of behavioural weight management interventions have been limited because few trials report relevant analyses and heterogeneity in the categorisation of inequality characteristics prevents meta-analysis. An individual participant data meta-analysis (IPD-MA) allows us to reanalyse all trials with available data in a uniform way. We aim to conduct an IPD meta-analysis of UK randomised controlled trials to examine whether there are inequalities in the attendance and effectiveness of behavioural weight interventions.

**Methods and analysis** In a recently published systematic review, we identified 17 UK-based randomised controlled trials of primary care-relevant behavioural interventions, conducted in adults living with overweight or obesity and reporting weight outcomes at baseline and 1-year follow-up. The corresponding author of each trial will be invited to contribute data to the IPD-MA. The outcomes of interest are weight at 12-months and intervention attendance (number of sessions offered vs number of sessions attended). We will primarily consider whether there is an interaction between intervention group and characteristics where inequalities occur, such as by gender/sex, socioeconomic status or age. The IPD-MA will be conducted in line with the Preferred Reporting Items for Systematic Reviews and Meta-analyses of IPD guidelines.

**Ethics and dissemination** No further ethical approval was required as ethical approval for each individual study was obtained by the original trial investigators from appropriate ethics committees. The completed IPD-MA will be disseminated at conferences, in a peer-reviewed journal and contribute to the lead author's PhD thesis. Investigators of each individual study included in the final IPD-MA will be invited to collaborate on any publications that arise from the project.

## STRENGTHS AND LIMITATIONS OF THIS STUDY

⇒ To the best of our knowledge, this will be the first individual participant data meta-analysis examining inequalities in the attendance and effectiveness of behavioural weight management interventions.
⇒ Individual participant data meta-analyses allow for data harmonisation, which increases the amount of data that can be used and the statistical power to explore whether inequalities are present in the attendance and outcome of behavioural weight management interventions.
⇒ Due to complexities in developing data sharing agreements and harmonisation of inequality variables in acquiring data from other countries, only UK-based trials are included in this individual participant data meta-analysis.

## INTRODUCTION

Inequalities in overweight and obesity are widely recognised—those who experience higher levels of socioeconomic deprivation, especially women, are more likely to live with obesity than those who more affluent.[1 2] Similarly, comorbidities of obesity are more common in those experiencing socioeconomic deprivation.[3] It has also been suggested that interventions focusing on individual behaviour change, such as behavioural weight management interventions, may exacerbate health inequalities.[4 5] These intervention-generated inequalities may occur at different stages, including intervention uptake, attendance and effectiveness, and across many individual characteristics that stratify health opportunities (such as access to healthcare) and outcomes. These characteristics are summarised by the PROGRESS-Plus framework: Place of Residence, Race/ethnicity, Occupation, Gender/sex, Education, Socioeconomic status, Social Capital, plus other

factors for which discrimination could occur such as age and sexual orientation.[6]

We recently conducted a systematic review to synthesise evidence on how different measures of inequality moderate the uptake, attendance and effectiveness of behavioural weight management interventions in adults.[7] We found that most trials did not consider whether inequalities were generated in the studied intervention; where these analyses were conducted, most found no evidence of inequalities. Where an inequalities gradient was observed, intervention uptake, adherence and attrition generally favoured those considered as 'more advantaged' (such as those who are white, with higher income or older). Due to substantial differences in the reporting of measures of inequality, together with the low level of reporting of analyses of inequalities, we were unable to perform a quantitative synthesis of the reported results. Hence, it is not possible to fully explore inequalities using aggregated data from published literature alone. This lack of reporting may have occurred because individual trials may not be large enough to detect an interaction between moderators such as socioeconomic status (SES) and the outcome; the trials are likely to have been designed to just detect an overall effect.

These limitations can be addressed in part by conducting a meta-analysis of individual participant data (IPD), which requires the central collation, aggregation and reanalysis of IPD from relevant trials.[8 9] This allows for data in each study to be analysed and defined in a uniform way, overcoming heterogeneity issues associated with using aggregate data. Meta-analysis of IPD may provide sufficient statistical power to consider whether there are inequalities in uptake, attendance and effectiveness of interventions.[8 10]

### Research questions

1. To what extent does the effectiveness of behavioural weight management interventions (defined as the difference in weight change between intervention and control) differ by individual characteristics that stratify health opportunities and outcomes (as defined using the PROGRESS-Plus Framework)?
2. To what extent do the weight outcomes of those who have participated in a behavioural weight management trial (defined as weight change in the overall cohort) differ by individual characteristics that stratify health opportunities and outcomes?
3. To what extent does attendance at behavioural weight management interventions differ by individual characteristics that stratify health opportunities and outcomes?

### METHODS AND ANALYSIS

This IPD meta-analysis responds to limitations identified in our previous systematic review on inequalities in the uptake of, adherence to and effectiveness of behavioural weight management interventions. The Preferred

Reporting Items for Systematic Reviews and Meta-Analyses IPD (PRISMA-IPD) extension will be followed when reporting this study.[11]

### Trial identification
#### Search strategy

This study includes UK-based trials of behavioural weight management interventions that we identified through a previous systematic review.[7] We focused on UK-based trials to reduce heterogeneity in measures of the PROGRESS-Plus characteristics and in the context in which the interventions were delivered. Characteristics such as ethnicity and socioeconomic status are conceptualised differently in different countries, which make synthesising data across these characteristics inappropriate or not possible. For example, socioeconomic status in the UK is often captured using Indices of Multiple Deprivation (IMD), an area-based measure, which is not replicated in other countries. There are also pragmatic reasons for focusing on UK-based trials; the complexity of arranging cross-country data sharing would have made the timelines for this project unviable.

Studies published since the search strategy in the systematic review was conducted, were identified through an updated Medline search and through discussions with the corresponding authors of the included trials. The inclusion and exclusion criteria we used to identify relevant trials for this IPD meta-analysis are:

1. Participants: adults aged 18 years and over with overweight or obesity (body mass index (BMI) $>25 \text{kg/m}^2$ with no upper limit) who were deemed suitable (either by the applicable study team or healthcare practitioner) for weight loss or weight loss maintenance intervention. Participants may have additional risk factors such as hypertension, dyslipidaemia, impaired glucose tolerance or impaired fasting glucose. Studies were excluded if the population was not selected based on a weight-related measure, included participants with $BMI<25 \text{kg/m}^2$, were selected based on having a chronic disease where weight loss is part of disease management or being pregnant, or if the intervention was targeted at parents to change behaviour of children. These exclusions were made to reduce heterogeneity between study populations and to be consistent with previous reviews.[7 12] Only studies conducted in the UK were eligible.
2. Interventions: behavioural weight management interventions with the primary aim of supporting weight loss or weight loss maintenance. Studies were included if they were conducted in, or were applicable to, primary care settings. Interventions may have been delivered alone or as part of a wider multicomponent intervention targeting diet and nutrition, physical activity, sedentary behaviour or a combination of these. Studies of pharmacological and surgical interventions were excluded unless the trial included behavioural only and control arms. Interventions were considered feasible for application to primary care if they were conduct-

ed in a healthcare setting or were widely available in the community at a national or regional level (such as commercial weight loss programmes, text message and other digital-based interventions); examples of settings that are not relevant to primary care include interventions delivered in inpatient settings, or in residential care homes.

3. Comparators: wait-list control, usual care or minimal intervention (such as generic print or electronic materials).
4. Outcomes: studies must report weight change in kg at a time point between the 12-month or 18-month follow-up.
5. Study designs: randomised or cluster randomised controlled trials (RCTs).

The trials we identified as being eligible for inclusion are outlined in online supplemental table 1.[13–29]

### Study outcomes, exposures and covariates
#### Outcomes
Outcomes are weight (kg) at 12-month follow-up and intervention attendance. Where data allow, attendance will be measured as the percentage of offered sessions which were attended.

#### Exposures
Exposure variables are selected measures of the PROGRESS-Plus criteria where data are likely to be available in UK-based trials of behavioural weight management interventions. Coding of each exposure variable will depend on the variables and coding used in each study providing IPD. Should the provided variables be suitably homogeneous, then we anticipate that the coding for the IPD meta-analysis will consist of the following:

► Ethnicity (coded using the England and Wales census categories: Asian or Asian British/black, African, Caribbean or black British/mixed or multiple ethnic groups/other ethnic group/white/Missing).
► Occupation (employed/unemployed/retired/other (ie, student, carer, voluntary work)/missing).
► Gender/sex (male/female/non-binary/not provided or other).
► Religion (coded using the England and Wales census categories: no religion/Christian/Buddhist/Hindu/Jewish/Muslim/Sikh/other religion/missing).
► Education (university degree or equivalent, or higher/postsecondary education/A-levels or equivalent/GCSEs or equivalent/no formal qualifications/missing).
► SES—household income (<£40 000/≥£40 000/missing).
► SES—IMD (1—most deprived/2/3/4/5—least deprived/missing).
► Social capital—relationship status (single/married or cohabiting/widowed, separated or divorced/missing).
► Age (continuous integer values).

► Disability (has physical-related or mental health-related disability/no disability/missing).
► Randomised group (control/intervention).

#### Covariates
► Baseline weight (kg).

### Risk of bias assessment
We will use Cochrane's risk of bias tool for RCTs (RoB 2) to assess the risk of bias in all studies meeting our inclusion criteria.[30] The tool facilitates researchers to consider bias across six domains: the randomisation process; allocation concealment; participant and trial personnel blinding; blinding of outcome assessment; incomplete outcome data; and selective reporting. A rating of 'low risk', 'high risk' or 'unclear' will be assigned to each domain by two contributors independently. Where disagreements occur, these will be resolved by discussion to reach consensus or through consultation with a third contributor. We will present the results of the risk of bias assessment in a summary figure outlining a study's overall risk of bias in addition to the risk of bias in each domain.

### Data collection and management
Our approach to collecting and aggregating the IPD was informed by the PRISMA-IPD extension and previously published IPD meta-analysis protocols.[8 11 31–33]

#### Invitation of authors
Trial investigators of all eligible trials were invited by email, using contact details acquired through trial publications, to contribute data and collaborate on this study. The email outlined our research aims and the specific data we were requesting.

#### Data collection
Standardised data specification forms will be sent to trial authors. Data will be requested in Microsoft Excel format; however, data will be accepted in any format. Once received, a master copy of each trial dataset will be saved in its original format and preserved. Any non-Microsoft Excel format datasets will be converted and then imported into Stata V.17 (StataCorp. 2021. Stata Statistical Software: Release 17.). We will also ask for detailed definitions of the measures used in the trial so we can ensure appropriate harmonisation.

#### Data checking
Once data are received from trial authors, they will be checked for quality and to ensure they pertain to the correct trial. Descriptive statistics (sample size, demographic variables, weight loss or BMI change) will be performed for each individual trial; should discrepancies occur between our analysis and the original trial publication, then the study authors will be contacted for clarification. Should clarification not be received, then the size of the discrepancy will be considered. If it is small and unlikely to bias the results, then the data will be included in the IPD meta-analysis. Large data inaccuracy

and excessive missing data (vs what is reported in the trial publication) may lead to a trial being excluded.

## Database creation and aggregation

A single database will be created containing data from all the trials. There is likely to be differences between studies in the coding of measures of inequalities. For example, there is variation in measures of SES that are used. Hence, once all data are received and the differences in coding across the measures become apparent, we will discuss among the core research team (JMB, MPK, SJG and AA) the best approach to achieve consistent reporting across all measures.

Once the data checking has been completed, variables will be recoded to match the coding of the IPD database. The data from each individual trial will then be copied into the IPD database and checked to ensure the integrity of the data has been maintained through the merge. Each individual trial dataset will be given a unique identifier prior to the merge.

## Trials where IPD are not available

Where we are unable to obtain IPD for an eligible trial, we will ask the trial investigator if they are able to conduct the analyses using the same coding of variables as defined in the Analysis of study outcomes section and provide us with the outcome statistics. We will offer this as an option to ensure that we are able to include as much relevant data as possible and we will provide the relevant code to facilitate this. If the outcome statistics are provided from any trials, we will meta-analyse this together with the trials for which we obtained IPD. If we synthesise results from data that we did not receive IPD for, we will conduct sensitivity analyses excluding these data. Further sensitivity analyses excluding studies with 'high' risk of bias will also be conducted.

## Statistical analysis

Due to our research questions exploring treatment effect and covariate interactions, we decided that a two-stage IPD meta-analysis would be most appropriate. In the first stage, regression analyses are performed individually in each trial. Then in the second stage combines the outcome estimates from each model using a standard meta-analysis approach (eg, random-effects meta-analysis).[34] We are using a two-stage approach because it inherently avoids aggregation bias and controls for trial-level confounding, to which one-stage IPD meta-analyses are more susceptible.[34] An additional benefit of performing a two-stage IPD meta-analysis is that trials for which we are unable to acquire individual-level data may still be included in the synthesis provided the relevant outcome statistics can be obtained. Data analysis will be conducted using Stata V.17 (StataCorp. 2021. Stata Statistical Software: Release 17, StataCorp).

## Baseline characteristics

We will describe the baseline characteristics for randomised group and each PROGRESS-Plus characteristic. This will be completed for each trial and as an overall aggregate of all participants included in the meta-analysis. We will compare these characteristics descriptively with data on the prevalence of obesity in the population, such as the Health Survey for England and other studies that have considered who routinely accesses behavioural weight management interventions.[35–37]

## Analysis of study outcomes

We will conduct six sets of analyses (if there are sufficient data for each outcome), two for each of our research questions, as we will synthesise data on weight loss interventions separately to data for weight loss maintenance interventions. As all outcomes of interest are continuous, we will use multivariable linear regression models, and include the relevant parameter estimates and standard errors from the models in random-effects meta-analyses. Heterogeneity will be assessed using $tau^2$, which summarises between-studies variance, and a 95% prediction interval which indicates the range in which 95% of the true effects lie. Inconsistency will be assessed using $I^2$, which indicates the proportion of total variability in the observed effects that is due to heterogeneity.

The subgroups used for each exposure variable are listed below (reference subgroup in bold). If free-text responses are available for any 'other' subgroup for each exposure, we will recode to the most appropriate subgroup in that exposure. If this is not possible, we will recode 'other' to missing. 'Prefer not to say' responses will also be recoded to missing. We anticipate that certain subgroups of some variables will likely have few, if any, data—in particular some subcategories of religion or relationship status. These will be recoded to missing and excluded from the analyses.

► Ethnicity (**white**/ethnic minorities (excluding White minorities)).
► Occupation (unemployed/**employed**/unable to work/retired/student).
► Gender/sex (**female**/male/other or non-binary).
► Religion (**no religion**/Christian/Buddhist/Hindu/Jewish/Muslim/Sikh/other religion).
► Education (no formal qualifications/**GCSEs, O-levels or equivalent**/A-levels or equivalent/some additional training/university degree).
► Index of Multiple Deprivation (**1—most deprived**/2/3/4/5—least deprived).
► Household income (**<£40 000**/≥£40 000).
► Relationship status (**single**/married or cohabiting/widowed, separated or divorced).
► Age (continuous).
► Randomised group (**control**/intervention).

We will present summary statistics for weight and attendance outcomes separately for each trial and as combined values across all trials.

*Research question 1: to what extent does the effectiveness of behavioural weight management interventions (defined as the difference in weight change between intervention and control) differ by individual characteristics that stratify health opportunities and outcomes (as defined using the PROGRESS-Plus Framework)?*

These analyses will focus on intervention effects by subgroup. Multivariable linear regression models will be used to test the null hypothesis that there is no interaction between each PROGRESS-Plus characteristic and intervention group on weight at 12 months. Each model will be adjusted for age and gender/sex (with the exception of the models where age and gender/sex are considered as the exposure variables) and baseline weight. The interaction terms will then be meta-analysed across trials.

*Research question 2: to what extent do the weight outcomes of those who have participated in a behavioural weight management trial (defined as weight change in the overall cohort) differ by individual characteristics that stratify health opportunities and outcomes?*

In these analyses, each trial will be analysed as a cohort study. Using multivariable regression, we will estimate the association between the 'exposure' variable (ie, each PROGRESS-Plus characteristic we have sufficient data for), and weight at 12-month follow-up. Each model will be adjusted for age and gender/sex (with the exception of the models where age and gender/sex are considered as the exposure variables), baseline weight and assigned intervention. Associations will be estimated for each exposure subgroup within a trial, and these associations will then be meta-analysed across trials.

*Research question 3: to what extent does attendance of behavioural weight management interventions differ by individual characteristics that stratify health opportunities and outcomes?*

For the third research question, each trial will be analysed as a cohort study. Attendance will be considered as a percentage—the number of sessions attended divided by the maximum possible number of sessions a participant could attend—and treated as a continuous variable. Multivariable regression models will be used to estimate the association between each PROGRESS-Plus characteristic and attendance. Each model will be adjusted for age and gender/sex (with the exception of the models where age and gender/sex are considered as the exposure variables). Associations will be estimated for each exposure subgroup within a trial, and these associations will then be meta-analysed across trials.

### Missing data
A complete-case analysis will be performed, that is, participants who have missing data for either the outcome, exposure or covariates will be excluded.

### Sensitivity analysis
As highlighted in the 'Trials where IPD are not available' section, if we synthesise results from data that we did not receive IPD for, we will conduct sensitivity analyses excluding these data. Sensitivity analyses excluding studies with 'high' risk of bias will also be conducted to consider whether these studies have an impact on the results. Should sufficient data be obtained, we will also conduct further analyses to consider whether intervention characteristics affect inequalities in attendance and effectiveness. These analyses may include comparisons by intervention length, digital versus non-digital and group based versus individually based.

### Patient and public involvement
As part of the protocol development for the preceding systematic review, we received comments on a lay summary from a patient and public involvement (PPI) representative on the project aims and our definition of the PROGRESS-Plus characteristics.[38] These aims and definitions have been brought forward into this IPD meta-analysis project. We will seek further PPI input on our harmonisation of the subgroups of the exposure variables to ensure our categorisations are appropriate, and a PPI representative will contribute to the interpretation of data and will coauthor the final manuscript.

## DISCUSSION
There is some evidence from our previously conducted systematic review that those we may consider to be 'more advantaged' (such as having more years of education, higher income, being white and being older) may be the most likely to maintain attendance to and have better outcomes from behavioural weight management interventions.[7] However, evidence was mixed and in that review we were unable to quantitatively synthesise data on attendance and weight outcomes due to heterogeneity in study populations (our review focused on all Organisation for Economic Co-operation and Development countries) and measures of the PROGRESS-Plus characteristics (eg, race and ethnicity are captured very differently in the USA vs the UK). This heterogeneity will be partly addressed in this IPD meta-analysis by focusing on trials from a single country (the UK) and through the data harmonisation that can be achieved when access to IPD is obtained.

This IPD meta-analysis will have several implications for public health policy, practice and research. The analyses may identify certain sociodemographic groups that have lesser attendance, or attain lesser weight loss. From a research perspective, future work could seek to establish why this may be the case; for public health policy, it is important to identify groups where interventions may be generating or exacerbating inequalities so additional support or provision can be offered to prevent this from occurring.

### Strengths and limitations
There are several strengths of conducting an IPD meta-analysis in comparison to a conventional meta-analysis. IPD meta-analyses are particularly useful for considering moderators of intervention outcomes,[8] due to the increased statistical power gained by pooling data

(although this is not guaranteed for all moderating variables, as it depends on the available data). Harmonisation of variables across studies means more data can be pooled together, leading to more robust analyses and conclusions. A further strength of IPD meta-analyses is that they go beyond published data, which may be limited in the measures reported. Receiving the original trial data also allows for increased data checking and increased validation of previously published results.[9]

However, there are also limitations of conducting an IPD meta-analysis. Even though the raw trial data will be acquired, analysis is dependent on the measures assessed in each original trial, and may be limited. Data harmonisation that is required to conduct an IPD meta-analysis may lead to some data being excluded from the analyses as it is unlikely to be possible to harmonise all data from different measures of each PROGRESS-Plus characteristic. A further limitation is that the estimates of inequality are influenced by the distribution of characteristics within each study. For example, studies with a narrow age range might not identify interactions between intervention effects and age. Finally, we are only looking at UK-based trials of behavioural weight management interventions, which may limit the generalisability of our findings to other countries or healthcare systems.

## Ethics and dissemination

Ethical approval was not required for this study as no primary data are to be collected, and the IPD are to be analysed in accordance to the purpose for which they were originally collected for. Ethical approval for each eligible trial for this IPD meta-analysis was obtained by the original investigators of each trial.

We anticipate that the completed IPD meta-analysis will be published in a scientific journal; one collaborator from each trial contributing IPD will be invited to be a coauthor on the publication. The findings from this IPD meta-analysis study may also be presented at relevant public health and obesity research conferences, and will contribute to the lead investigator's PhD thesis.

**Contributors** JMB conceived and designed the study, developed the analysis plan and drafted the manuscript. JM contributed to study design and the analysis plan, and reviewed drafts of the manuscript. SS contributed to the analysis plan and reviewed drafts of the manuscript. JL, MPK and SJG contributed to study design and reviewed drafts of the manuscript. AA conceived and designed the study, contributed to the analysis plan and reviewed drafts of the manuscript. All authors approved the final version for publication.

**Funding** JMB, AA, SJG and SS are supported by the Medical Research Council (MRC) (Grant MC_UU_00006/6). The University of Cambridge has received salary support in respect of SJG from the National Health Service in the East of England through the Clinical Academic Reserve. This work is funded by UKRI grant MC_UU_00006/6.

**Competing interests** JM is a trustee for the Association for the Study of Obesity (unpaid role). JL has received research support from Slimming World and consulting fees from Novo Nordisk and is an employee of AstraZeneca. This work was performed before she became an AstraZeneca employee and AstraZeneca had no role in the work. MPK has undertaken consultancy for Slimming World and led the obesity and weight management guidelines development for NICE from 2005 until 2014. SJG is principal investigator on a publicly funded (NIHR) trial in which the intervention is provided by WW (formerly Weight Watchers) at no cost. AA is principal investigator on two publicly funded (NIHR, MRC) trials where the intervention is provided by WW (formerly Weight Watchers) at no cost.

**Patient and public involvement** Patients and/or the public were involved in the design, or conduct, or reporting, or dissemination plans of this research. Refer to the Methods section for further details.

**Patient consent for publication** Not applicable.

**Provenance and peer review** Not commissioned; externally peer reviewed.

**ORCID iDs**
Jack M Birch http://orcid.org/0000-0001-6292-1647
Julia Mueller http://orcid.org/0000-0002-4939-7112
Jennifer Logue http://orcid.org/0000-0001-9549-2738

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
