## [Reviewer comments · BMJ Open]

ARTICLE DETAILS

TITLE (PROVISIONAL)	Are there inequalities in the attendance and effectiveness of behavioural weight management interventions for adults in the UK? Protocol for an individual participant data (IPD) meta-analysis
AUTHORS	Birch, Jack; Mueller, Julia; Sharp, Stephen; Logue, Jennifer; Kelly, Michael; Griffin, Simon; Ahern, Amy

VERSION 1 – REVIEW

REVIEWER	Madigan, Claire Loughborough University
REVIEW RETURNED	13-Sep-2022

GENERAL COMMENTS	This is a great study and I look forward to reading the results. I have a few minor comments that they authors might want to consider. 1. In the abstract the authors state that one of the inequalities might be sex, however in the methods gender is referred to. I can imagine the older trials would be referring to sex so the team might want to be specific about what they are measuring and why.2. The authors refer to in the research questions about “stratify health opportunities and outcomes” It is not clear what is meant by this and is this additional part even needed. I would consider removing this or define health opportunities and which variables they are in reference to.3. Line 25, page 6 – this is a bit open with the “may include, but not limited to” - I am not sure it is needed. Perhaps you could say instead a behavioural weight management intervention with the aim to help people lose weight or maintain weight loss.4. Could the authors describe why they have excluded participants that are using weight loss as part of disease management? From a practical perspective we want to help people to lose weight to manage their condition. This means you could also include the trial by Lean et al: https://pubmed.ncbi.nlm.nih.gov/29221645/5. Could the authors explain application to primary care (line 3, page 7) does this mean the intervention could be referred from primary care. This would be particularly important for the weight loss maintenance interventions as in particular LIMIT and NU level were recruited through the community with no primary care involvement. Or perhaps remove the primary care part and put in community setting with exclusions of hospital outpatients. Therefore you would not have to include tier 3 programmes.6. In the risk of bias section how are you going to define missing outcome data – if it is greater than 80% is that high and is there any rule for difference between comparator and intervention groups? Also are you only including studies with objective weights as otherwise you might consider self-report weight as high risk of bias in measuring outcome data.7. Page 12, line 19 -could you give an example as it was not clear
--

	which variables would likely have few? 8. Is religion yes or no and what is the reference group? 9. What is the reference group for relationship status? 10. I found it difficult to understand what the difference was between research question 1 and research question 2. As the effectiveness of behavioural outcomes is measured by the amount of weight people lose. Both the questions are examining weight loss by individual characteristics. Could you make it clearer what is the difference and what each one tells you. 11. Table 1: The Daley trial was called LIMIT and consider adding an author to the Norfolk Diabetes Prevention study for consistency.
--	---

REVIEWER	Albert, Steven M. University of Pittsburgh, Behavioral and Community Health Sciences
REVIEW RETURNED	18-Oct-2022

GENERAL COMMENTS	This research synthesis will be valuable for providing an individual participant data meta-analysis to examine inequalities in the attendance and effectiveness of behavioural weight management interventions. The use of the PROGRESS-Plus correlates and two-stage IPD meta-analysis are appropriate. Some areas to clarify include the following:  1. Research questions 1 and 2 seem quite similar. The only difference appears to be use of an interaction term in assessing question 2. Perhaps the second could be subsumed within the first. 2. Research question 3 is limited to correlates of attendance. Yet the trials have other implementation outcomes that may be relevant, such as submission of weekly logs of activity, records of food intake, and weekly weigh-in compliance. The authors could consider including these as well for studies reporting this information.
---

VERSION 1 – AUTHOR RESPONSE

Reviewer: 1

Claire Madigan, Loughborough University Comments to the Author:

This is a great study and I look forward to reading the results. I have a few minor comments that they authors might want to consider.

1. In the abstract the authors state that one of the inequalities might be sex, however in the methods gender is referred to. I can imagine the older trials would be referring to sex so the team might want to be specific about what they are measuring and why.

The reviewer is correct that some studies report gender and some report sex. While we recognise that these variables have distinct meaning, they are often used interchangeably in the literature. Indeed, some studies refer to sex in some papers and gender in others from the same data set and it is not clear the precise question that was asked. We have now relabelled this variable Gender/Sex to reflect that this is a harmonised variable.

2. The authors refer to in the research questions about “stratify health opportunities and outcomes” It is not clear what is meant by this and is this additional part even needed. I would consider removing this or define health opportunities and which variables they are in reference to. To make it clear what is meant by characteristics that health opportunities and outcomes, we have added the below 2 sentences into the first paragraph of the introduction (page 4, lines 8-12)

“These intervention-generated inequalities may occur at different stages, including intervention uptake, attendance and effectiveness, and across many individual characteristics that stratify health

opportunities (such as access to healthcare) and outcomes. These characteristics are summarised by the PROGRESS-Plus. framework: Place of Residence, Race/ethnicity, Occupation, Gender/sex, Education, Socioeconomic status, Social Capital, plus other factors for which discrimination could occur such as age and sexual orientation.”

3. Line 25, page 6 – this is a bit open with the “may include, but not limited to” - I am not sure it is needed. Perhaps you could say instead a behavioural weight management intervention with the aim to help people lose weight or maintain weight loss.

We have now removed this sentence, as it does not add further context to the information about eligible interventions already included in the paragraph.

4. Could the authors describe why they have excluded participants that are using weight loss as part of disease management? From a practical perspective we want to help people to lose weight to manage their condition. This means you could also include the trial by Lean et al:

We excluded studies that are using weight loss as part of disease management for pragmatic and scientific reasons. Pragmatically, we have kept the inclusion criteria the same as the USPSTF (<https://jamanetwork.com/journals/jama/fullarticle/2702877>) and our own systematic review (<https://doi.org/10.1111/obr.13438>) so we could extract the relevant studies from our systematic review as well as performing an updated search. Scientifically, we decided against including trials where weight loss has been used as part of disease management as these studies have different eligibility criteria and this could have increased heterogeneity. It is conceivable that intervention effectiveness could differ in populations with specific illnesses, and that intervention uptake, adherence and effectiveness could be patterned differently in populations seeking weight loss for disease management.

5. Could the authors explain application to primary care (line 3, page 7) does this mean the intervention could be referred from primary care. This would be particularly important for the weight loss maintenance interventions as in particular LIMIT and NU level were recruited through the community with no primary care involvement. Or perhaps remove the primary care part and put in community setting with exclusions of hospital outpatients. Therefore you would not have to include tier 3 programmes.

“Conducted in, or were applicable to, primary care settings” – this text highlights that to be eligible for inclusion in the individual participant data meta-analysis, then the intervention would need to be delivered in, referred from, primary care or be a model of intervention that could feasibly be delivered in or referred from primary care in future. This means that not all interventions eligible for inclusion needed to have been delivered in primary care. This is consistent with the terminology used in the USPSTF report and our previous systematic review.

6. In the risk of bias section how are you going to define missing outcome data – if it is greater than 80% is that high and is there any rule for difference between comparator and intervention groups? Also are you only including studies with objective weights as otherwise you might consider self-report weight as high risk of bias in measuring outcome data.

In the guidance document for completing the RoB 2 tool (available from <https://sites.google.com/site/riskofbiastool/welcome/rob-2-0-tool/current-version-of-rob-2?authuser=0>), it suggests that percentage missingness of outcome data should not be used alone in assigning high risk of bias to the incomplete outcome data domain. We will consider percentage of missingness alongside whether there are demographic differences between those with and without outcome data when assigning a risk of bias.

The inclusion criteria do not exclude studies that use self-reported weight data as the outcome. With the RoB 2 tool, where the outcome assessor is aware of the intervention received (which self-report participants would be) and the reporting of this outcome could be biased by this knowledge, it would indicate a high risk of bias. So if any studies included in our individual participant data meta-analysis used self-reported weight as the outcome, they would likely receive a ‘high risk’ rating for the blinding of outcome assessment domain.

7. Page 12, line 19 -could you give an example as it was not clear which variables would likely have few?

We have added in some examples to improve clarity, the sentence now reads “We anticipate that certain subgroups of some variables will likely have few , if any, data – in particular some sub-categories of religion or relationship status.” (page 12, line 14-15)

8. Is religion yes or no and what is the reference group?

We have now added in the recoded categories for religion, and made clear ‘No religion’ is the reference group. (page 12, line 19)

9. What is the reference group for relationship status?

We have now made the reference group for relationship status (Single) bold in text. (page 12, line 24)

10. I found it difficult to understand what the difference was between research question 1 and research question 2. As the effectiveness of behavioural outcomes is measured by the amount of weight people lose. Both the questions are examining weight loss by individual characteristics. Could you make it clearer what is the difference and what each one tells you.

We have changed the wording of research questions 1 and 2, as well as the order, to provide extra clarity. The now first research question reads (page 5, lines 9-11):

1. To what extent does the effectiveness of behavioural weight management interventions (as defined by the difference in weight change between intervention and control) differ by individual characteristics that stratify health opportunities and outcomes?

The additional wording in brackets makes clear that this research question is focused on if there are inequalities in the difference in weight change between the intervention group and to the control group. We have made this question one as this is the primary measure of inequality in intervention effectiveness. The second research now reads:

2. To what extent do the weight outcomes of those who have participated in a behavioural weight management trial (defined as weight change in the overall cohort) differ by individual characteristics that stratify health opportunities and outcomes (as defined using the PROGRESS-Plus Framework)?

This makes clearer that the analyses for this research question look at the overall sample as a cohort. This question provides additional context for considering if inequalities resulting from intervention effects exacerbate (or reduce) existing inequalities in weight loss in those motivated to lose weight or generates new inequalities.

11. Table 1: The Daley trial was called LIMIT and consider adding an author to the Norfolk Diabetes Prevention study for consistency.

We have added the LIMIT trial name to Table 1. As explained above, we have now removed reference to the Norfolk Diabetes Prevention study as it does not meet our inclusion criteria.

Reviewer: 2

Dr. Steven M. Albert, University of Pittsburgh Comments to the Author:

This research synthesis will be valuable for providing an individual participant data meta-analysis to examine inequalities in the attendance and effectiveness of behavioural weight management interventions. The use of the PROGRESS-Plus correlates and two-stage IPD meta-analysis are appropriate. Some areas to clarify include the following:

1. Research questions 1 and 2 seem quite similar. The only difference appears to be use of an interaction term in assessing question 2. Perhaps the second could be subsumed within the first. We have changed the wording of research questions 1 and 2, as well as the order, to provide extra clarity. The now first research question reads (page 5, lines 9-11):

1. To what extent does the effectiveness of behavioural weight management interventions (as defined by the difference in weight change between intervention and control) differ by individual characteristics that stratify health opportunities and outcomes?

The additional wording in brackets makes clear that this research question is focused on if there are inequalities in the difference in weight change between the intervention group and to the control group. We have made this question one as this is the primary measure of inequality in intervention effectiveness. The second research now reads:

2. To what extent do the weight outcomes of those who have participated in a behavioural weight management trial (defined as weight change in the overall cohort) differ by individual characteristics that stratify health opportunities and outcomes (as defined using the PROGRESS-Plus Framework)?

This makes clearer that the analyses for this research question look at the overall sample as a cohort. This question provides additional context for considering if inequalities resulting from intervention effects exacerbate (or reduce) existing inequalities in weight loss in those motivated to lose weight or generates new inequalities.

2. Research question 3 is limited to correlates of attendance. Yet the trials have other implementation outcomes that may be relevant, such as submission of weekly logs of activity, records of food intake, and weekly weigh-in compliance. The authors could consider including these as well for studies reporting this information.

We did consider asking for data on other measures of intervention adherence, such as those mentioned by the reviewer. However, few studies reported such measures and there was substantial heterogeneity in measures amongst those that did. The most commonly collected measure of intervention adherence across our included studies was attendance. This increases the likelihood of successfully harmonising data on this outcome across the studies to produce a meaningful estimate of differences across the PROGRESS-Plus criteria.

VERSION 2 – REVIEW

REVIEWER	Albert, Steven M. University of Pittsburgh, Behavioral and Community Health Sciences
REVIEW RETURNED	25-Jan-2023
GENERAL COMMENTS	The authors have responded well to reviewer concerns. One linger issue may be differences in age groups across the intervention studies and its effect on estimates of disparities.

VERSION 2 – AUTHOR RESPONSE

Reviewer: 2

Dr. Steven M. Albert, University of Pittsburgh Comments to the Author:

The authors have responded well to reviewer concerns. One linger issue may be differences in age groups across the intervention studies and its effect on estimates of disparities.

[Editor: Please consider commenting on this issue in the manuscript]

Thank you to Dr Albert for this suggestion. We agree this is a limitation of this study, and have added the following text into the strengths and limitations section (lines 7-10, page 16):

“A further limitation is that the estimates of inequality are influenced by the distribution of characteristics within each study. For example, studies with a narrow age range might not identify interactions between intervention effects and age.”